# A Mechanism of Isoorientin-Induced Apoptosis and Migration Inhibition in Gastric Cancer AGS Cells

**DOI:** 10.3390/ph15121541

**Published:** 2022-12-12

**Authors:** Tong Zhang, Yun-Hong Xiu, Hui Xue, Yan-Nan Li, Jing-Long Cao, Wen-Shuang Hou, Jian Liu, Yu-He Cui, Ting Xu, Ying Wang, Cheng-Hao Jin

**Affiliations:** 1College of Life Science & Technology, Heilongjiang Bayi Agricultural University, Daqing 163319, China; 2Hemodialysis Center, Daqing Oilfield General Hospital, Daqing 163001, China; 3National Coarse Cereals Engineering Research Center, Daqing 163319, China; 4College of Food Science & Technology, Heilongjiang Bayi Agricultural University, Daqing 163319, China

**Keywords:** isoorientin, gastric cancer, cell apoptosis, cell cycle, cell migration, reactive oxygen species

## Abstract

Isoorientin (ISO) is a flavonoid compound containing a luteolin structure, which can induce autophagy in some tumor cells. This study investigated the impact of ISO in gastric cancer AGS cells, and performed an experimental analysis on the main signaling pathways and transduction pathways it regulates. CCK–8 assay results showed that ISO reduced the survival rate of gastric cancer AGS cells, but the toxicity to normal cells was minimal. Hoechst 33342/PI double staining assay results showed that ISO induced apoptosis in gastric cancer AGS cells. Further analysis by flow cytometry and Western blot showed that ISO induced apoptosis via a mitochondria-dependent pathway. In addition, the level of reactive oxygen species (ROS) in gastric cancer AGS cells also increased with the extension of the ISO treatment time. However, cell apoptosis was inhibited by preconditioning cells with N–acetylcysteine (NAC). Moreover, ISO arrested the cell cycle at the G2/M phase by increasing intracellular ROS levels. Cell migration assay results showed that ISO inhibited cell migration by inhibiting the expression of p–AKT, p–GSK–3β, and β–catenin and was also related to the accumulation of ROS. These results suggest that ISO-induced cell apoptosis by ROS–mediated MAPK/STAT3/NF–κB signaling pathways inhibited cell migration by regulating the AKT/GSK–3β/β–catenin signaling pathway in gastric cancer AGS cells.

## 1. Introduction

Globally, gastric cancer ranks third in terms of prevalence and death rates after lung cancer [1,2]. In China, gastric cancer is more common in people over 50 years of age [3,4]. In the present day, gastric cancer can be treated primarily through surgery and chemotherapy. Chemotherapy drugs, as the primary means of treatment after surgery, have an inhibitory effect on cancer cells, but they often destroy healthy normal cells, leading to the disorder of immune metabolism in the human body, with great side effects on the body [5,6]. There is, therefore, a need for a drug that is highly efficient and has a low risk of side effects. There are a number of natural products that have potential therapeutic effects on cancer cells while causing few side effects on other cells, so they have been subjected to intense research [7,8,9].

Isoorientin (ISO) is a flavonoid compound widely found in *Polygonum orientale* L., Patrinia, *Phyllostachys edulis* leaf, passionflower, and other plants [10,11,12]. ISO possesses a range of pharmacological actions, including anti–inflammatory, antiviral, and antibacterial activity, as well as weakening the development of liver fibrosis [13,14,15,16]. In addition, studies have shown that ISO has superior anticancer activity against breast cancer cells, hepatoblastoma, and other tumor cells [17,18,19,20]. The anticancer mechanism of ISO on stomach cancer cells, however, is unknown. In this study, ISO was examined for its effects on the apoptosis, proliferation, and migration of stomach cancer cells.

Apoptosis induction is currently one of the main ways to treat cancer cells. A high level of reactive oxygen species (ROS) promotes mitochondrial membrane potential reversal by activating mitogen–activated protein kinase (MAPK), leading to cytochrome c release and caspase–3 activation, ultimately resulting in apoptosis [21,22]. In addition, ROS can also regulate the STAT3 and NF–κB signaling pathways and cooperate with the MAPK signaling pathway to promote cell apoptosis [23].

By contrast, inhibiting tumor cell growth and metastasis play a critical role in cancer treatment. Glycogen synthase kinase 3β (GSK–3β) is the main intracellular serine/threonine kinase and has a bidirectional regulatory effect on the growth and development of tumor cells. When it is inhibited as a cancer–promoting factor, it promotes β–catenin entry into the nucleus, thereby activating the β–catenin signaling pathway and inhibiting the spread of tumor cells [24].

The effect of ISO on the apoptosis and migration of gastric cancer AGS cells was explored in this study, and its probable mechanism was found.

## 2. Results

### 2.1. Isoorientin Reduces the Viabilities of Gastric Cancer Cells

As shown in Figure 1A,B, ISO reduced the viability of twelve typical gastric cancer cell lines in a dose-dependent manner with a toxic effect higher than 5–FU, and ISO had a lower cytotoxic effect on normal cells. The IC_50_ values of each cell treated for this drug are summarized (Table 1). As shown in Figure 1C,D, ISO produced similar results in a time–dependent manner at a specific dose. Among the above gastric cancer cells, with an IC_50_ value of 36.54 µM, AGS cells had the highest sensitivity to ISO.

### 2.2. Isoorientin Induces AGS Cell Apoptosis

As shown in Figure 2A,B, the edge of the AGS cells shrank after treatment with ISO. At 24 h, fluorescence intensity reached its maximum, which was higher than that of the 5–FU treatment group over the same period. Figure 2C illustrates the findings of flow cytometry, which demonstrated that the apoptosis of AGS cells treated with ISO increased over time and reached 35.76% of apoptotic cells at 24 h. As shown in Figure 2D, the AGS cells’ mitochondrial membrane potential fluctuation trend was inversely associated with the length of time they underwent ISO treatment. As seen in Figure 2E, ISO markedly lowered the protein expression level of Bcl–2 while dramatically increasing the protein expression levels of Bad, cyto–c, cle–caspase–3, and cle–PARP. The mitochondrial apoptotic pathway was involved in ISO–induced AGS cell apoptosis, according to our findings.

### 2.3. Isoorientin Regulates MAPK/STAT3/NF–κB Signaling Pathways

As seen in Figure 3A, ISO considerably decreased the expression levels of p–ERK, p–STAT3, and NF–κB while significantly upregulating the expression levels of p–JNK, p–p38, and IκBα in a time-dependent way. Then, we tested several nuclear proteins and found that ISO significantly reduced the levels of STAT3, NF–κB, and p–IκBα (Figure 3B). As seen in Figure 3C–E, compared with the ISO alone treatment groups, FR180204, SP600125, and SB203580 inhibited the expression of p–ERK, p–JNK, and p–p38 in AGS cells, respectively. Meanwhile, pretreatment with ERK inhibitors enhanced the decrease in p–STAT3 and Bcl–2 protein expression levels and inhibited the increase in cle–caspase–3 protein expression, while the level of p–STAT3 after pretreatment with the JNK inhibitor or p38 inhibitor was higher than that of ISO treatment alone. The above results indicate that the MAPK–dominated ISO–induced AGS cell apoptosis is an upstream signaling pathway.

### 2.4. Isoorientin Regulates the MAPK/STAT3/NF–κB Signaling Pathways That Are Mediated by ROS

As shown in Figure 4A, in a time-dependent way, ISO dramatically increased the accumulation of ROS in AGS cells, reaching 83.42% after 24 h of treatment. However, ISO reduced ROS accumulation in normal gastric GES–1 cells (Figure 4B). As shown in Figure 4C, ISO–induced AGS cell apoptosis was inhibited by NAC treatment. As shown in Figure 4D,E, the expression levels of p–JNK, p–p38, IκBα, cle–caspase–3, and cle–PARP regulated by ISO was decreased by NAC treatment. ISO-regulated expression levels of p–ERK, p–STAT3, NF–κB, nuclear protein STAT3, NF–κB, and p–IκBα were increased by NAC treatment. These results suggest that ISO induces AGS cell apoptosis by promoting ROS accumulation.

### 2.5. Isoorientin Arrested the AGS Cell Cycle at the G2/M Phase

As seen in Figure 5A,B, following ISO treatment, fewer cells were in the G0/G1 phase (down from 64.64% to 47.69%), but more cells were in the G2/M phase. ISO decreased expression levels of p–AKT, CDK1/2, and Cyclin B, and increased expression levels of p21 and p27. As seen in Figure 5C,D, when NAC was added before ISO treatment, the number of G0/G1 cells increased from 61.76% to 70.12%, and ISO–induced expression of the above proteins was similarly reversed. The above results indicate that ISO could arrest the AGS cell cycle in the G2/M phase by up–regulating ROS.

### 2.6. Isoorientin Inhibited AGS Cell Migration via the AKT/GSK–3β/β–catenin Signaling Pathways

As seen in Figure 6A,B, ISO reduced the AGS cell migration area and the number of AGS cells, demonstrating that ISO had the ability to inhibit the migration of AGS cells. As seen in Figure 6C,D, we further detected migration-related proteins and found that the expression levels of p–AKT, p–GSK–3β, Twist, ZEB1, N–cadherin, and β–catenin were all decreased, while the expression levels of E-cadherin were increased. These protein expression levels were reversed after NAC pretreatment. The above results indicate that ISO inhibited AGS cell migration via the ROS–mediated AKT/GSK–3β/β–catenin signaling pathways.

## 3. Discussion

Numerous studies have revealed that Chinese herbal medicines can replace certain chemicals due to their mild effects and low toxicity. In terms of inhibiting tumor growth and inducing tumor cell apoptosis, higher concentrations of traditional Chinese medicine extracts have been shown to have good biological activities [25,26,27]. Isoorientin (ISO), a flavonoid compound, has been demonstrated to decrease growth and induce apoptosis in some tumor cells. In the preliminary experiment of this study, it was found that under the same conditions, the killing effect of 5–FU on most of the gastric cancer cells from 12 different sources was better than that of other chemotherapy drugs. So, we selected 5–FU as the positive control in our subsequent research on ISO. In this study, ISO was used to treat 12 kinds of gastric cancer cells, and the analysis showed that ISO had excellent killing effects on these gastric cancer cells, and the effect was better than 5–FU. While inhibiting the growth of tumor cells, ISO was significantly less harmful to normal cells than 5–FU in the control group (Figure 1).

In this study, the CCK–8 assay results show that the effect of ISO on AGS cells was the most significant among these gastric cancer cells. AGS cells came from human gastric adenocarcinoma epithelial cell lines. Some studies have shown that gastric cancer cells from different sources have different biomarkers, different cell metabolites, and different drug sensitivity. In addition, the occurrence of gastric cancer is not only related to genetic susceptibility but also related to genetic mutations. Both KRAS and CTNNB1 genes were mutated in AGS cells. KRAS regulates downstream effectors of the MAPK pathway through phosphorylation. CTNNB1 can encode β–catenin and play a role in regulating cell migration [28]. In this study, we found that ISO can induce the apoptosis of AGS cells through the MAPK pathway and inhibit the migration of AGS cells by regulating β–catenin. We speculate that the origin of AGS cells and the mutated genes may be the reason why AGS cells are most affected by ISO. AGS cells were finally selected in this study to prove the anticancer effect of ISO.

In the process of apoptosis, Bcl–2 family proteins dominate the changes in mitochondrial membrane permeability [19,29]. This study has shown that ISO can reduce the expression of Bcl–2 and increase the expression of Bax by blocking the PI3K/Akt pathway, thus inducing the apoptosis of HepG2 cells [30]. Similarly, this study found that ISO up–regulated the expression of pro–apoptotic protein Bad and down–regulated the expression of anti-apoptotic protein Bcl–2, resulting in a decrease in the mitochondrial membrane potential and ultimately leading to AGS cell apoptosis (Figure 2). We found that when the ISO treatment time of cells was more than 24 h or close to 48 h, the cells basically lost their original form and were not enough for research. Therefore, when exploring the anti–tumor effects of ISO on AGS cells, 24 h of ISO treatment was selected as the last time point. In addition, in order to confirm the molecular mechanism of ISO–induced AGS cell apoptosis. Through the analysis of signal pathways, this study found that in the ISO–induced apoptosis of AGS cells, the ISO–regulated MAPK signaling pathway and STAT3 signaling pathway, together with the NF–κB signaling pathway, play an anti–cancer role (Figure 3).

Flavonoids are often used as antioxidants because of their ability to scavenge oxygen–free radicals [31]. However, it has been shown that their antioxidant effects depend on the cells being treated [32]. This study has shown that the ISO treatment of HepG2 cells significantly increased intracellular ROS levels, which were 33.58% higher than normal liver cells HL–7702 [33]. In tumor and normal cell lines of the same organ, different concentrations of one flavonoid are commonly observed to induce oxidative stress and have antioxidant effects. However, our aim was to explore the effects of ISO on ROS in different cell types. Therefore, we analyzed the regulatory effects of ISO on ROS levels in gastric cancer cells and normal cells under the same ISO treatment concentration. This study found that with the same ISO treatment concentration, decreased ROS accumulation in normal gastric GES–1 cells as ISO treatment time increased, but ROS accumulation in AGS cells continued to increase. This study also found that ISO induced AGS cell apoptosis by the up–regulation of ROS, activated p38 and JNK signaling pathways, and inhibited ERK, STAT3, and NF–κB signaling pathways (Figure 4).

When a cell becomes cancerous, the host loses control of the cell cycle. Therefore, the principle of cell cycle regulation can be used in tumor treatment to block the cycling of cancer cells, thereby preventing the metastasis and spread of cancer cells and, at the same time, inducing tumor cell apoptosis [34]. RT–PCR analysis showed that ISO significantly reduced gene transcription and Cyclin D, Cyclin E, and CDK 2 expression levels in HepG2 cells [19]. This study analyzed the cycle of AGS cells after ISO treatment by flow cytometry. It was found that ISO led to the G2/M cycle arrest of AGS cells through the accumulation of ROS (Figure 5).

The development of cancer is usually accompanied by rapid spread and invasion. Studies have shown that after the ISO treatment of pancreatic cancer cells, the expression levels of VEGF, MMP2, and MMP9 decreased, but the expression level of E–cadherin increased [35]. However, this study discovered that ISO inhibits AGS cell migration via modulating the AKT/GSK–3β/β–catenin signaling pathway and through regulating ROS generation (Figure 6).

In summary, through the ROS–mediated MAPK/STAT3/NF–κB and AKT signaling pathways, ISO triggered apoptosis and cell cycle arrest in AGS cells. ISO, on the other hand, reduced cell migration via ROS–mediated GSK–3β signaling pathways (Figure 7). Furthermore, ISO has the advantage of being less toxic to normal cells. Therefore, this study speculates that ISO could be a promising medication for the treatment of AGS.

## 4. Materials and Methods

### 4.1. Cell Culture

Human gastric cancer cells and human normal lung cells were obtained from the American Type Culture Collection (ATCC, Manassas, VA, USA). Sage Biotechnology Co., Ltd. (Shanghai, China) provided the human normal gastric cells, liver cells, and renal cells. 10% FBS, 100 U/mL penicillin, and 100 µg/mL streptomycin in RPMI 1640 culture medium (Gibco, Waltham, MA, USA) was used to culture a variety of cells, including gastric cancer cells (AGS, KATO–3, MKN–28, MKN–45, SNU–5, SNU–216, SNU–484, SNU–668), normal liver cells (L–02), and normal lung cells (IMR–90). Other gastric cancer cells, normal gastric cells (GES–1), and normal kidney cells (293T) were cultured in DMEM (Gibco) with the same contents. In sterile cell culture, all cells were cultured at 37 °C and 5% CO_2_ in a SanYo incubator (Osaka, Japan).

### 4.2. Cell Viability Analyses

Four types of normal cells and twelve types of gastric cancer cells were seeded into 96–well plates (1 × 10^4^ cells/well) and cultured overnight. They were treated with ISO and 5–FU (MedChem Express, Princeton, NJ, USA) at different concentrations for 24 h. The viability of the cells was determined using the Cell Counting Kit–8 (Solarbio, Beijing, China). Each cell was treated with 10 µL of a CCK-8 reagent for 3 h. Then, the results of the Microplate readers (BioTek Instruments Inc., Winooski, VT, USA) were used to calculate the half maximal inhibitory concentration (IC_50_) value of the above-mentioned cells. Cell viability was determined by treating the cells with ISO and 5–FU IC_50_ values (0, 3, 6, 12, 24, and 36 h).

### 4.3. Cell Apoptosis Analysis

AGS cells were plated in 6–well dishes (1 × 10^5^ cells/well) and cultured overnight. The cells were treated with 36.54 µM ISO and 36.54 µM 5–FU (the IC_50_ value of AGS cells) for 0, 3, 6, 12, and 24 h. Cell apoptosis was analyzed using the Apoptosis and Necrosis Assay Kit (Beyotime, Shanghai, China). A total of 200 µL of cell staining buffer, 3 µL of Hoechst, and 2 µL of PI were added. Cells were observed using the cell imaging system (Thermo Fisher Scientific, Shanghai, China). In addition, 100 μL of an Annexin V-binding buffer, 4 μL of Annexin V–FITC, and 3 μL PI were added to ISO–treated AGS cells, and flow cytometry (Beckman Coulter, Brea, CA, USA), and the cell imaging system was used.

### 4.4. Mitochondrial Membrane Potential Analysis

AGS cells were seeded in 3.5 cm Petri dishes (1 × 10^5^ cells/dish) and cultured overnight. For 0, 3, 6, 12, and 24 h, the cells were exposed to 36.54 µM ISO. The JC–1 Assay Kit was used to assess the mitochondrial membrane potential of AGS cells (Solarbio). After centrifugation, AGS cells were collected and incubated with 1 mL JC–1 at 37 °C for 30 min. Then, AGS cells were suspended in a 1 mL binding solution two times and were detected by flow cytometry.

### 4.5. Extraction of Nucleoproteins

AGS cells were seeded into 3.5 cm Petri dishes and treated with 36.54 µM ISO for 0, 3, 6, 12, and 24 h. Samples were prepared with a Nuclear Protein Extraction Kit (Solarbio). Centrifugation was performed at 500 g for 2 min to separate the supernatant from the precipitation and was preserved for later use. A total of 100 mL of plasma protein extract was added to the precipitate and incubated with ice. After oscillation, they were centrifuged at 14,000 g at 4°C for 15 min, and 80 µL of nuclear protein extract was added to the precipitation. The supernatant was centrifuged again under the same conditions to prepare the sample.

### 4.6. Western Blot Analysis

AGS cells were seeded into 6 cm Petri dishes (6 × 10^5^ cells/dish) and treated with 36.54 µM ISO for 0, 3, 6, 12, and 24 h. A 30 min centrifugation at 12,000 rpm and 4 °C was followed by mixing the supernatants with a 5× buffer and boiling them for five minutes. The OD value was measured at 595 nm with a spectrophotometer, and the protein concentrations were normalized. Proteins (20 µL per sample) were separated by 8–12% SDS–PAGE gel electrophoresis and were transferred to a nitrocellulose filtration membrane (NC membrane) and incubated with 5% skim milk for 2 h. The primary antibody was incubated overnight at 4 °C on the NC membrane. Santa Cruz Biotechnology (Dallas, TX, USA) provided all the primary antibodies. The primary antibodies used were Bad (1:1500; cat. no. sc–493), Bcl–2 (1:1500; cat. no. sc–7382), cyto–c (1:2000; cat. no. sc–13156), cle–caspase–3 (1:1500; cat. no. sc–373730), cle–PARP (1:1500; cat. no. sc–8007), α–tubulin (1:2500; cat. no. sc–47778), p–ERK (1:1500; cat. no. sc–7383), ERK (1:1000; cat. no. sc–154), p–JNK (1:1500; cat. no. sc–6254), JNK (1:1500; cat. no. sc–7345), p–p38 (1:1500; cat. no. sc–7973), p38 (1:1500; cat. no. sc–7149), p–STAT3 (1:1500; cat. no. sc–8059), STAT3 (1:1500; cat. no. sc–8019), NF–κB (1:1500; cat. no. sc–8008), IκBα (1:1500; cat. no. sc–1643), p–IκBα (1:1500; cat. no. sc–8404), Lamin B1 (1:2500; cat. no. sc–374015), p–AKT (1:1000; cat. no. sc–7985–R), AKT (1:1000; cat. no. sc–8312), CDK1/2 (1:1500; cat. no. sc–53219), Cyclin B (1:1500; cat. no. sc–245), p21 (1:1500; cat. no. sc–397), p27 (1:1500; cat. no. sc–528), p–GSK–3β (1:1000; cat. no. sc–373800), GSK–3β (1:1500; cat. no. sc–377213), Twist (1:1500; cat. no. sc–81417), ZEB1 (1:1000; cat. no. sc–515797), E–cadherin (1:1000; cat. no. sc–8426), N–cadherin (1:1000; cat. no. sc–59987), and β–catenin (1:1000; cat. no. sc–7963). Secondary antibodies (ZSGB–bio, Beijing, China) labeled with horseradish peroxidase were then added, and the NC blots were incubated at room temperature for 2 h before being combined with the enhanced chemiluminescence (ECL) substrate (Thermo Fisher Scientific). An Amersham Imager 600 (GE Healthcare, Beijing, China) was used to detect changes in protein expression. Protein relative densities were calculated using the Image J program. The endogenous controls were α-tubulin and Lamin B1. In addition, the protein content was determined by pretreatment with ERK inhibitors (FR180204, 10 µM), JNK inhibitors (SP600125, 10 µM), p38 inhibitors (SB203580, 10 µM) (MCE, Middlesex, NJ, USA), or reactive oxygen scavengers NAC (10 mM) (Beyotime) for 30 min prior to ISO treatment.

### 4.7. Measurement of ROS Generation

AGS cells and GES–1 cells were seeded in 3.5 cm Petri dishes and treated with 36.54 µM ISO for 0, 3, 6, 12, and 24 h, respectively. ROS levels in the AGS and GES–1 cells were measured using the ROS Assay Kit (Beyotime). The AGS cells were spun at 8000 rpm for 5 min before being treated with 10 µM of DCFH–DA fluorescent probe for 30 min. ROS accumulation in AGS and GES–1 cells was measured using flow cytometry. N–acetyl–L–cysteine (NAC, Beyotime) was given to AGS cells for 30 min before ISO treatment to assess the ROS accumulation in AGS cells.

### 4.8. Cell Cycle Analysis

AGS cells were plated in 3.5 cm dishes and treated with 36.54 µM ISO for 0, 3, 6, 12, and 24 h. The DNA Content Detection Kit (Solarbio) was used to examine the cell cycle. The AGS cells were collected and fixed in 70% alcohol overnight. The cell samples were incubated with 100 µL RNase for 30 min at 37 °C to eliminate RNA and were then stained with 400 µL of PI for 30 min at 4 °C and were detected by flow cytometry.

### 4.9. Cell Migration Analyses

AGS cells were plated in 6–well dishes and used a 10 µL pipette tip to perform a vertical swipe when the cells were confluent. After careful washing with PBS, 36.54 µM ISO was used to treat the AGS cells, and the cell migration at each moment was observed using an Auto Cell Imaging System (MSHOT, Guangzhou, China). Image J was used to calculate the migratory areas of the cells. In addition, AGS cells were plated in the upper chamber of the 6–well Transwell plate. A total of 36.54 µM ISO was used to treat the AGS cells; 0.1% crystal violet staining solution was added and washed with PBS. The number of cell migrations was observed and counted under a microscope.

### 4.10. Statistical Analysis

Three separate experiments were run, and the results are presented as the mean ± SD. GraphPad Prism 5.0 was used to determine the IC_50_ of ISO. Tukey’s post hoc test was performed using SPSS version 21.0, and multiple comparisons across the groups were conducted using a one-way analysis of variance. Statistically significant differences are shown by the symbols * *p* < 0.05, ** *p* < 0.01, and *** *p* < 0.001.

## Figures and Tables

**Figure 1 pharmaceuticals-15-01541-f001:**
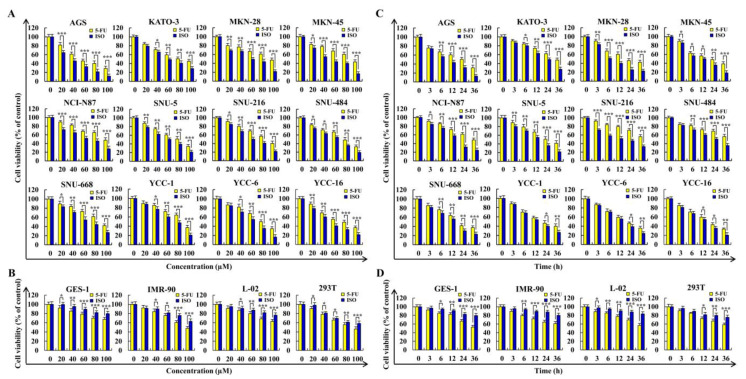
The cytotoxic effect of ISO. The CCK-8 assay was used to determine if cells had survived after being exposed to various doses or time of 5–FU or ISO. (**A**) Survival rate of gastric cancer cells. (**B**) Survival rate of normal cells. (**C**) Survival rate of gastric cancer cells. (**D**) Survival rate of normal cells. * *p* < 0.05, ** *p* < 0.01 and *** *p* < 0.001 vs. 5–FU.

**Figure 2 pharmaceuticals-15-01541-f002:**
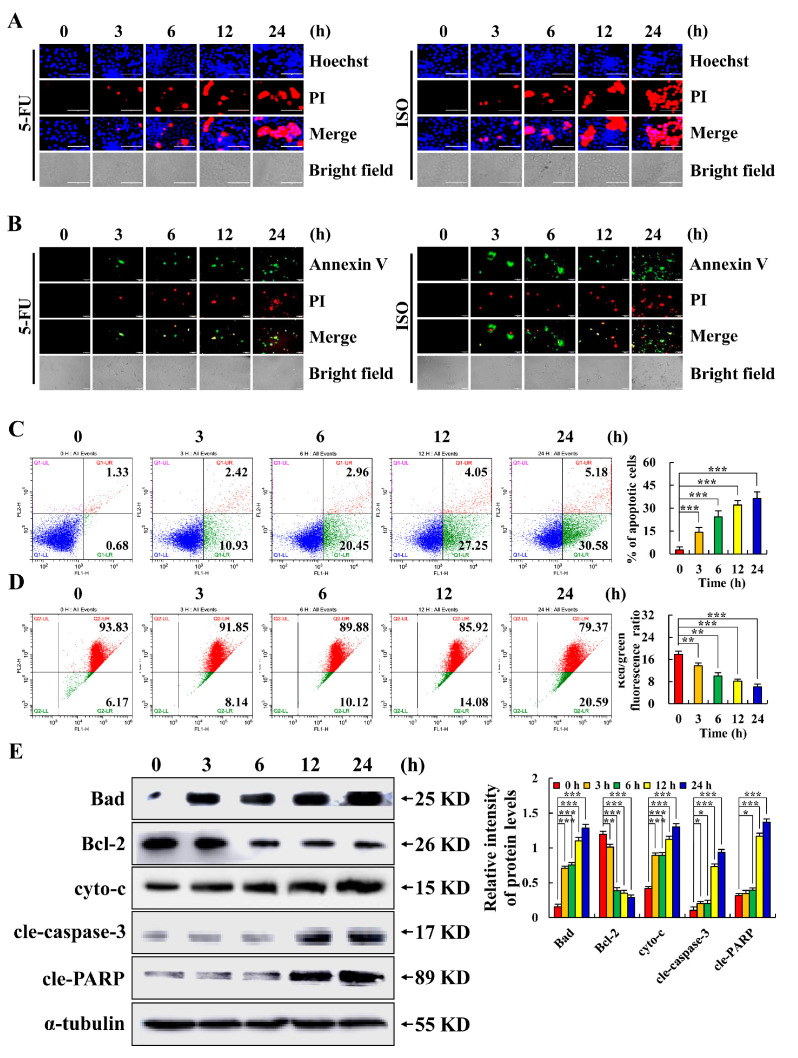
Apoptotic effects of ISO in AGS cells. A total of 36.54 µM ISO was applied to AGS cells from 0 to 24 h. (**A**) AGS cells after Hoechst and PI staining were observed by fluorescence microscope (original magnification, ×200). (**B**) AGS cells after Annexin V and PI staining were observed by fluorescence microscope (original magnification, ×400). (**C**) Apoptosis rate of AGS cells. (**D**) Flow cytometry was used to calculate the mitochondrial membrane potential. (**E**) Expression levels of apoptosis–related proteins in AGS cells treated with ISO. The α–tubulin was used as an internal reference protein. * *p* < 0.05, ** *p* < 0.01 and *** *p* < 0.001 vs. 0 h.

**Figure 3 pharmaceuticals-15-01541-f003:**
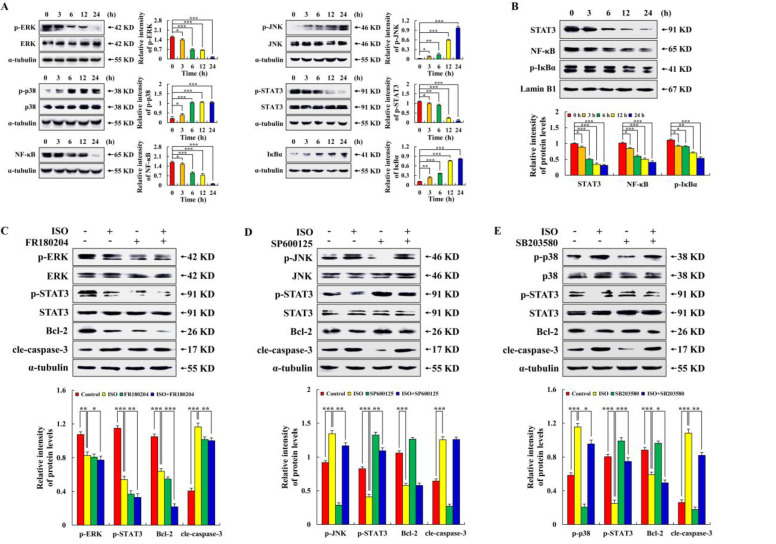
Regulation of ISO on the MAPK/STAT3/NF–κB signaling pathway in AGS cells. A total of 36.54 µM ISO was applied to AGS cells from 0 to 24 h. (**A**) MAPK/STAT3/NF–κB signaling pathway related protein expression level. (**B**) Nuclear protein expression level. (**C**) Expression levels of p–ERK, p–STAT3, Bcl–2, and cle–caspase–3 proteins in AGS cells treated with ISO (36.54 µM) and/or an ERK inhibitor (10 µM). (**D**) Expression levels of p–JNK, p–STAT3, Bcl–2, and cle–caspase–3 proteins in AGS cells treated with ISO (36.54 µM) and/or a JNK inhibitor (10 µM). (**E**) Expression levels of p–p38, p–STAT3, Bcl–2 and cle–caspase–3 proteins in AGS cells treated with ISO (36.54 µM) and/or a p38 inhibitor (10 µM). Lamin B1 and α–tubulin were used as an internal reference protein. * *p* < 0.05, ** *p* < 0.01 and *** *p* < 0.001 vs. 0 h or ISO + MAPK inhibition.

**Figure 4 pharmaceuticals-15-01541-f004:**
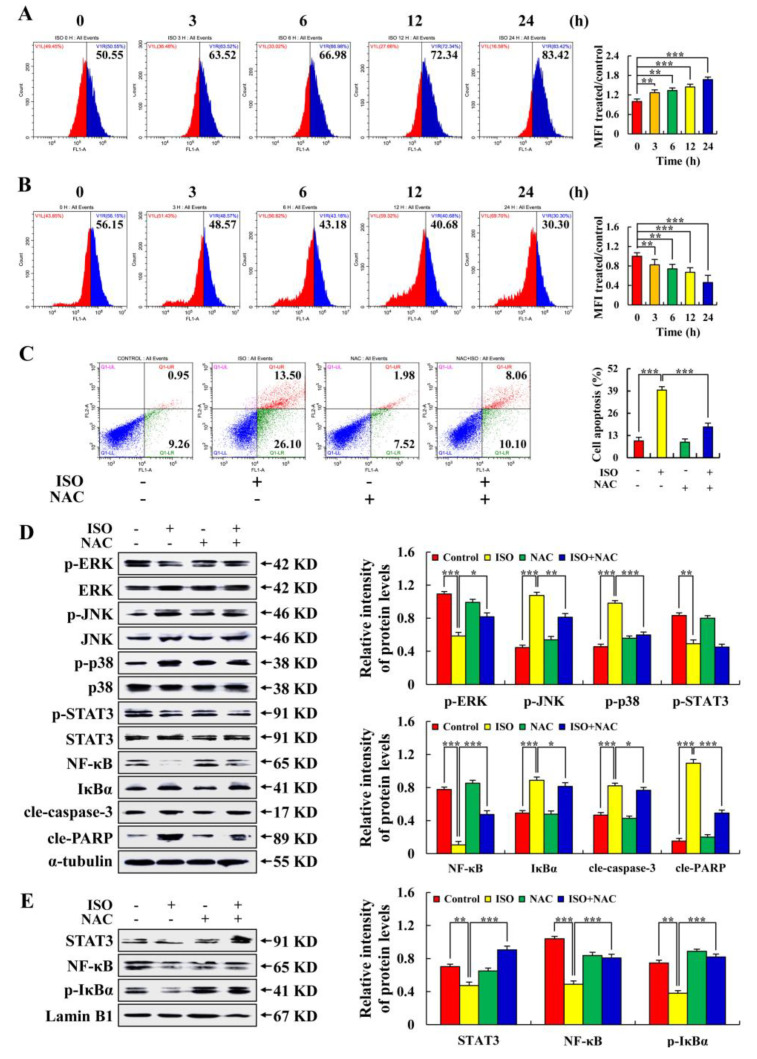
The promoting effects of ISO on ROS levels in AGS cells. A total of 36.54 µM ISO was applied to cells from 0 to 24 h. (**A**) AGS cell levels of ROS. (**B**) GES–1 cell levels of ROS. (**C**) Apoptosis rate of AGS cells treated with ISO (36.54 µM) and/or NAC (10 mM). (**D**) Expression levels of related signaling pathway proteins in AGS cells treated with ISO (36.54 µM) and/or NAC (10 mM). (**E**) Expression levels of nuclear proteins in AGS cells treated with ISO (36.54 µM) and/or NAC (10 mM). Lamin B1 and α–tubulin were used as an internal reference protein. * *p* < 0.05, ** *p* < 0.01 and *** *p* < 0.001 vs. 0 h or NAC + ISO.

**Figure 5 pharmaceuticals-15-01541-f005:**
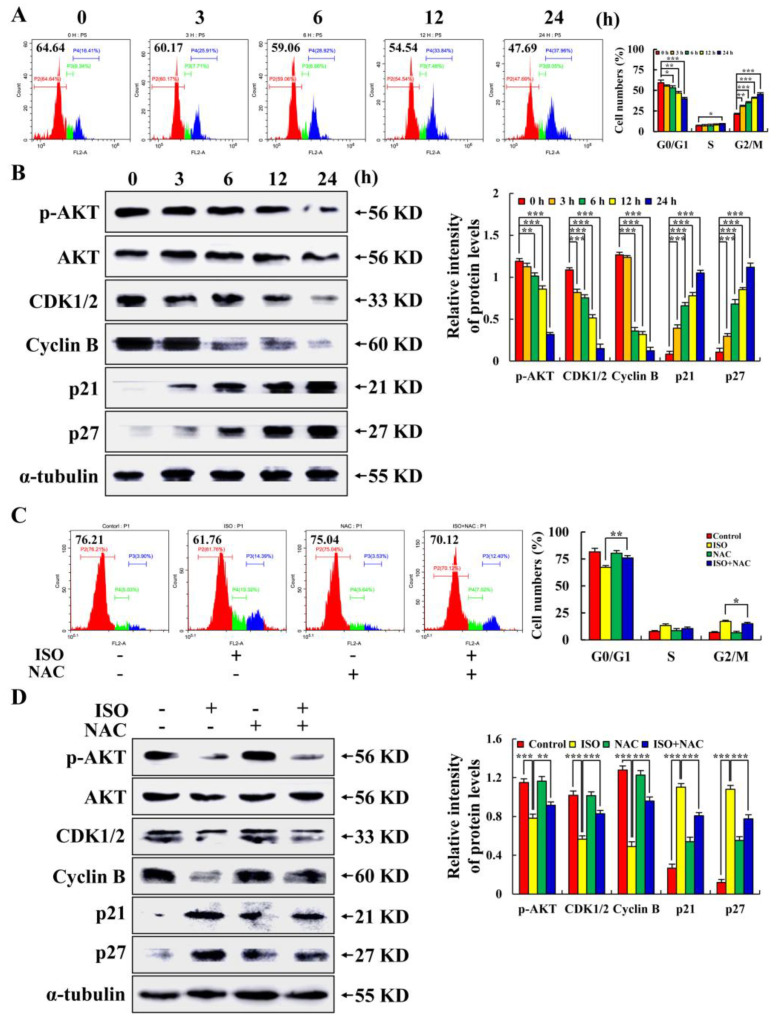
The arresting effects of ISO on the cell cycle in AGS cells. A total of 36.54 µM ISO was applied to AGS cells from 0 to 24 h. (**A**) Percentage of cell cycle number of AGS cells. (**B**) Expression of proteins involved in the G2/M cell cycle. (**C**) Percentage of cell cycle number of AGS cells treated with ISO (36.54 µM) and/or NAC (10 mM). (**D**) Expression of proteins involved in the G2/M cell cycle of AGS cells treated with ISO (36.54 µM) and/or NAC (10 mM), and α–tubulin was used as an internal reference protein. * *p* < 0.05, ** *p* < 0.01 and *** *p* < 0.001 vs. 0 h or NAC + ISO.

**Figure 6 pharmaceuticals-15-01541-f006:**
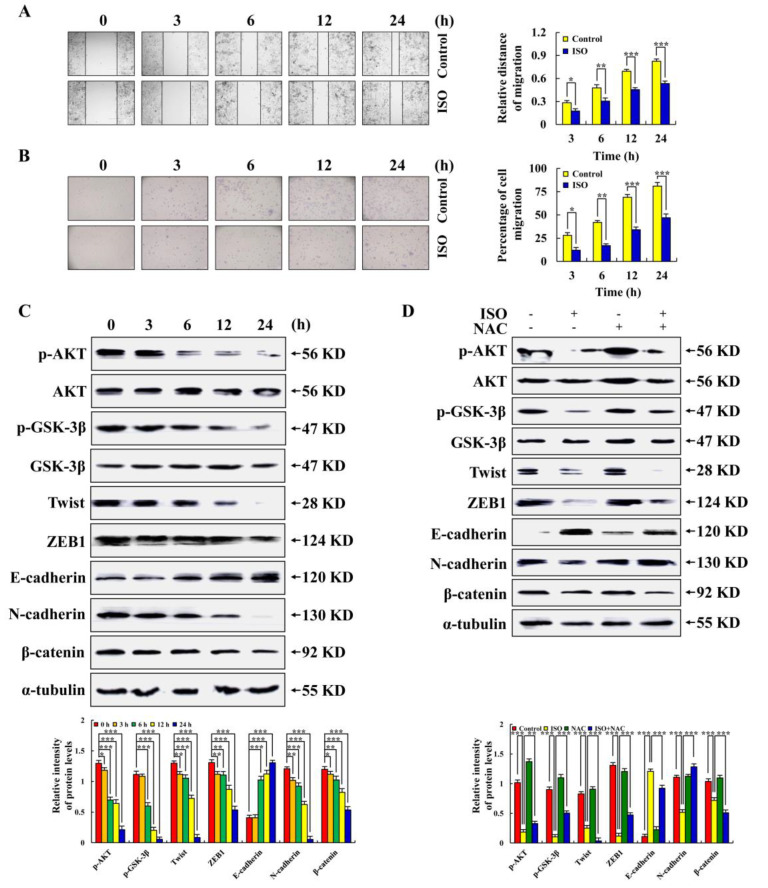
The inhibiting effects of ISO on migration in AGS cells. A total of 36.54 µM ISO was applied to AGS cells from 0 to 24 h. (**A**) Migration changes in AGS cells were observed by fluorescence microscope (original magnification, ×200). (**B**) The number of AGS cell migration was observed by fluorescence microscope (original magnification ×400). (**C**) Migration related protein expression level. (**D**) Expression of migration related proteins of AGS cells treated with ISO (36.54 µM) and/or NAC (10 mM), and α–tubulin was used as an internal reference protein. * *p* < 0.05, ** *p* < 0.01 and *** *p* < 0.001 vs. 0 h or NAC + ISO.

**Figure 7 pharmaceuticals-15-01541-f007:**
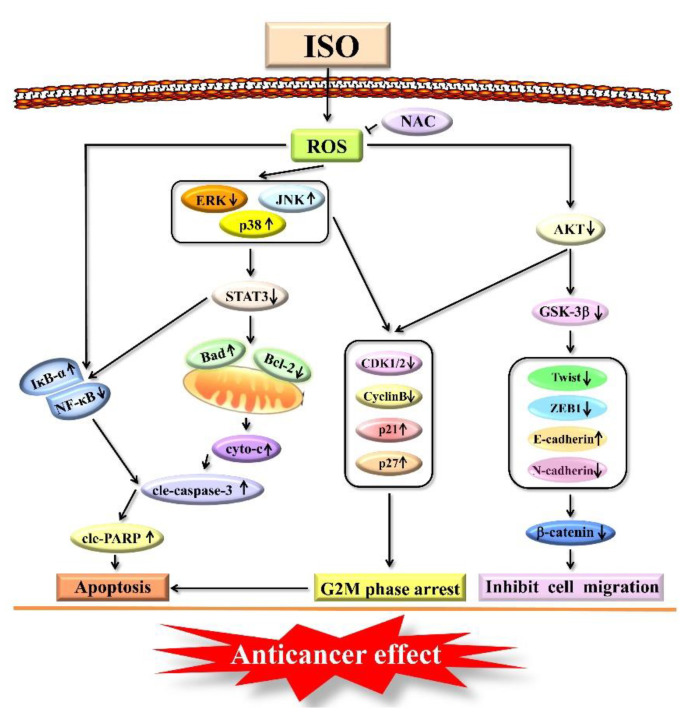
Schematic diagram of the signaling pathway of ISO in AGS cells to exert anti-cancer effects.

**Table 1 pharmaceuticals-15-01541-t001:** IC_50_ values of ISO and 5–FU in gastric cancer cells.

Cell Line	5–FU (µM)	ISO (µM)
AGS	56.27 ± 2.04	36.54 ± 1.93
KATO-3	81.11 ± 1.28	60.28 ± 1.21
MKN-28	98.12 ± 1.87	61.31 ± 2.42
MKN-45	89.67 ± 2.26	49.77 ± 2.18
NCI-N87	99.37 ± 1.98	68.16 ± 2.38
SNU-5	80.97 ± 2.01	62.21 ± 1.73
SNU-216	86.35 ± 2.84	59.57 ± 1.66
SNU-484	78.38 ± 1.76	65.59 ± 2.44
SNU-668	90.63 ± 1.57	71.74 ± 2.75
YCC-1	91.37 ± 1.03	78.62 ± 1.24
YCC-6	82.47 ± 1.63	66.83 ± 2.68
YCC-16	80.16 ± 2.72	56.72 ± 2.62

## Data Availability

Data is contained within the article.

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
