# Peer review of "A Mechanism of Isoorientin-Induced Apoptosis and Migration Inhibition in Gastric Cancer AGS Cells"

_pharmaceuticals, 2022, doi:10.3390/ph15121541_

Round 1

Reviewer 1 Report

Isoorientin can induce autophagy in some tumor cells. This study investigated the impact of ISO in gastric cancer Cells (AGS cells), the experimental analysis of the main signaling pathways and transduction pathways it regulated. Overall, the article is well written, there are a large number of results to support the results of the research, the paper is worth publishing, but small changes are recommended.

1.       Please explain the name of AGS-cells, whether gastric cancer cells (AGS-cells), this should be made clear in the abstract and introduction sections;

2.       The arrangement of the figures and the table must be on the same page, as shown in Table 1;

3.       The words or numbers in the figures should be clearly;

4.       The layout should not leave a lot of space, such as Page 3, 5;

5.       When citing references, the brackets should be empty;

6.       Some references do not have abbreviations for journal names.

Reviewer 2 Report

The manuscript submitted by Zhang et.al., tried to investigate the potential mechanisms through which Isoorientin exhibits its anticancer effect on AGS cell lines. The study is interesting and well-designed and written, but the following points should be addressed:

1.       Why did the authors choose 5-FU for their study and not other chemotherapies (cisplatin or Taxols as an example)? The justification should be included in the discussion

2.       It is very well known that protein expression is usually affected with time, so why did the authors use 24 hours as the last time point? It is recommended to do 24,48 and 72 hours time points, the justification should be included in the discussion

3.       Why AGS was the most affected cell line with ISO? This should be discussed in the discussion and related to the genetic signature of AGS compared to the other tested gastric cancer cell lines.

4.       It is common to see the same flavonoids induce oxidative stress (ROS) and have an antioxidant effect in tumor and normal cell lines of the same organ, but this is usually observed at different concentrations. The authors should discuss their findings regarding the effect of ISO on ROS on both GES and AGS cell lines at the same concentration ?!!!.

5.       The discussion is very brief and shallow, many findings have not been discussed or discussed superficially, the discussion should be revised and improved, and many of the above-mentioned points should be addressed and discussed in this section among others.

6.        In the materials and methods, the authors should include all the antibodies used and their sources.

7.       The figures are of a very poor resolution, the staining results for example are too difficult to look at. All figures should be enhanced.

Round 2

Reviewer 2 Report

the authors thankfully addressed all my comments and suggestions